# Lunar primitive mantle olivine returned by Chang'e-6

Si-Zhang Sheng[1], Shui-Jiong Wang [1,2] ✉, Qiu-Li Li [3], Shitou Wu [3], Hao Wang [3], Jun-Xiang Hua[1], Zhenyu Chen[4], Jin-Hua Hao[5], Bo Zhang[6], Yongsheng He[1,2] & Jian-Ming Zhu [1,2]

The lunar mantle is important for unraveling the Moon's formation and early differentiation processes. Here, we identify primitive lunar olivines in soils returned by the Chang'e-6 mission. These olivines have oxygen isotopic compositions plotting along the terrestrial fractionation line, and are characterized by high forsterite contents up to 95.6, and a broad range of nickel abundances from zero to 682 ppm. While the low-nickel (zero to 251 ppm), forsteritic olivines align with a Mg-suite origin, the most primitive, high-nickel olivines (337 to 682 ppm) have a different origin. They could be either the first olivine crystallized from the Lunar Magma Ocean (LMO) with an Earth-like initial composition, or crystallized from a hitherto unrecognized ultra-magnesian lava produced by extensive melting of the early LMO cumulate. The exposure of these mantle olivines was facilitated by their entrainment in ascending high-Mg lavas and conveyed to the surface at the South Pole-Aitken Basin.

Constraining the nature of the lunar mantle is fundamental to understanding the Moon's formation, differentiation and evolution[1]. Lunar scientists have long been searching for mantle materials ever since the first samples were brought back by the Apollo mission in year of 1969. A possible piece of the lunar mantle, as represented by a dunite fragment, has recently been identified in a lunar meteorite[2]. However, no definitive confirmed mantle materials have been found from the sample-return missions[3,4]. Our understanding of the ancient and modern lunar mantle is largely based on the physio-chemical models, laboratory experiments, geophysical and remote sensing observations, and coordinated analyses of volcanic products[5–17]. These sources of knowledge require further validation and are likely to be revised once the lunar mantle materials become accessible for direct analyses.

The Lunar Magma Ocean (LMO) paradigm posits a stratified mantle, where the lower mantle is dominated by magnesian olivine cumulates and transitions into a more ferroan orthopyroxene-dominated upper mantle[6,18]. However, as the Moon evolved, this initial cumulate stratigraphy was disrupted: the less dense olivine cumulates would have ascended upon the overlying high-density mantle via gravitational restructuring[19]. There is currently no consensus on what lithologies of the lunar upper mantle would have been[1,5,20].

The South Pole-Aitken Basin (SPA) on the Moon's farside is believed to be the most promising location for mantle materials to be exposed on the lunar surface[1]. On June 25, 2024, the Chang'e-6 (CE-6) mission successfully retrieved the first sample from the SPA (41.625 °S, 153.978 °W). Here, we identify highly magnesian olivines (Fo from 86.8 to 95.6; Fo = molar Mg/(Mg + Fe) * 100) in the CE-6 soils, and provide elemental and oxygen isotopic evidence that the nickel (Ni)-poor, forsteritic olivines are of Mg-suite origin, whereas the most primitive, Ni-rich olivines

[1]State Key Laboratory of Geological Processes and Mineral Resources, China University of Geosciences (Beijing), Beijing, China. [2]Frontiers Science Center for Deep-time Digital Earth, China University of Geosciences (Beijing), Beijing, China. [3]State Key Laboratory of Lithospheric and Environmental Coevolution, Institute of Geology and Geophysics, Chinese Academy of Sciences, Beijing, China. [4]MNR Key Laboratory of Metallogeny and Mineral Assessment, Institute of Mineral Resource, Chinese Academy of Geological Sciences, Beijing, China. [5]Institute of Earth Sciences, China University of Geosciences (Beijing), Beijing, China. [6]The Key Laboratory of Orogenic Belts and Crustal Evolution, School of Earth and Space Sciences, Peking University, Beijing, China. ✉e-mail: wsj@cugb.edu.cn

are likely of mantle origin and compositionally similar to the first olivines crystallized from the LMO.

## Results and discussion

Olivine grains are generally rare in the CE-6 soils[21]. By employing Tescan Integrated Mineral Analysis (TIMA) techniques, we identified 63 olivine grains in approximately 12 milligrams of CE-6 soils (CE6C0400YJFM005) (Fig. 1 and Supplementary Fig. 1). Half of the forsteritic olivine grains either occur as single anhedral mineral fragments, or are embedded together with anorthite in impact glasses (Fig. 1a–c and Supplementary Fig. 1). One particular lithic fragment (approximately 100 μm in width and 250 μm in length) contains numerous anhedral to euhedral olivines of approximately 5–50 μm in size in a groundmass composed of plagioclase, olivine, and spinel (Fig. 1d). The mineral phases in the groundmass are unfortunately too fine-grained for accurate analysis. The bulk compositions of the groundmass, obtained by electron microprobe analysis (EPMA, with a spot size of 6 μm) and semi-quantitative scanning electron microscopy (SEM) analysis, differ significantly from the impact glasses in the CE-6 soils but show similarities to the proposed parental melts of Mg-suite troctolite (Supplementary Fig. 2a; Supplementary Data 1)[22–26]. The fine-grained and spinifex-like texture of the groundmass suggests that it is a Mg-rich lava that underwent rapid cooling, equivalent to the extrusive

Mg-suite magmatism[27,28]. The high temperature expected for such extrusive Mg-rich lavas would be capable of assimilating the anorthositic crust and producing Mg-Al spinel[23,29–31]. Crystal fractionation simulations applied to the groundmass successfully reproduce the Mg-suite trend, with a crystallization sequence of dunite, troctolite to norite (Supplementary Fig. 2b, c).

The micro-olivines in the groundmass are generally <3 um in size and exhibit a narrow Fo range of 91.6 ± 1.2 (n = 9, 1sd) (Figs. 1, 2), which seems in Fe-Mg equilibrium with the melts of the groundmass (Supplementary Fig. 3). Large olivine crystals generally display skeletal and erosional morphologies, consisting of a wide compositional interior zone (Fo = 89.7–95.6) and a narrow overgrowth rim (mostly <1 μm in width) with Fo = 91.6 ± 0.9 (n = 8, 1sd), which is similar to the micro-olivines in the groundmass. The olivine rim is too narrow to obtain accurate data. Three olivine populations were distinguished based on the texture and chemical composition (Fig. 2 and Supplementary Fig. 4): Group I olivine is represented by the largest polygonal crystal (~50 μm) with a dissolution feature at the margin. Its interior is homogenous, with relatively low Fo (89.9–90.8) and low Ni (0–55 ppm); Group II olivine is predominantly hopper-shaped, containing an irregular, relic core featured by low Fo (89.7–92.4) and low Ni (0–149 ppm), and a more magnesian mantle with higher Fo (92.9–95.4) but similar Ni contents (8–188 ppm); Group III olivine is polyhedral to

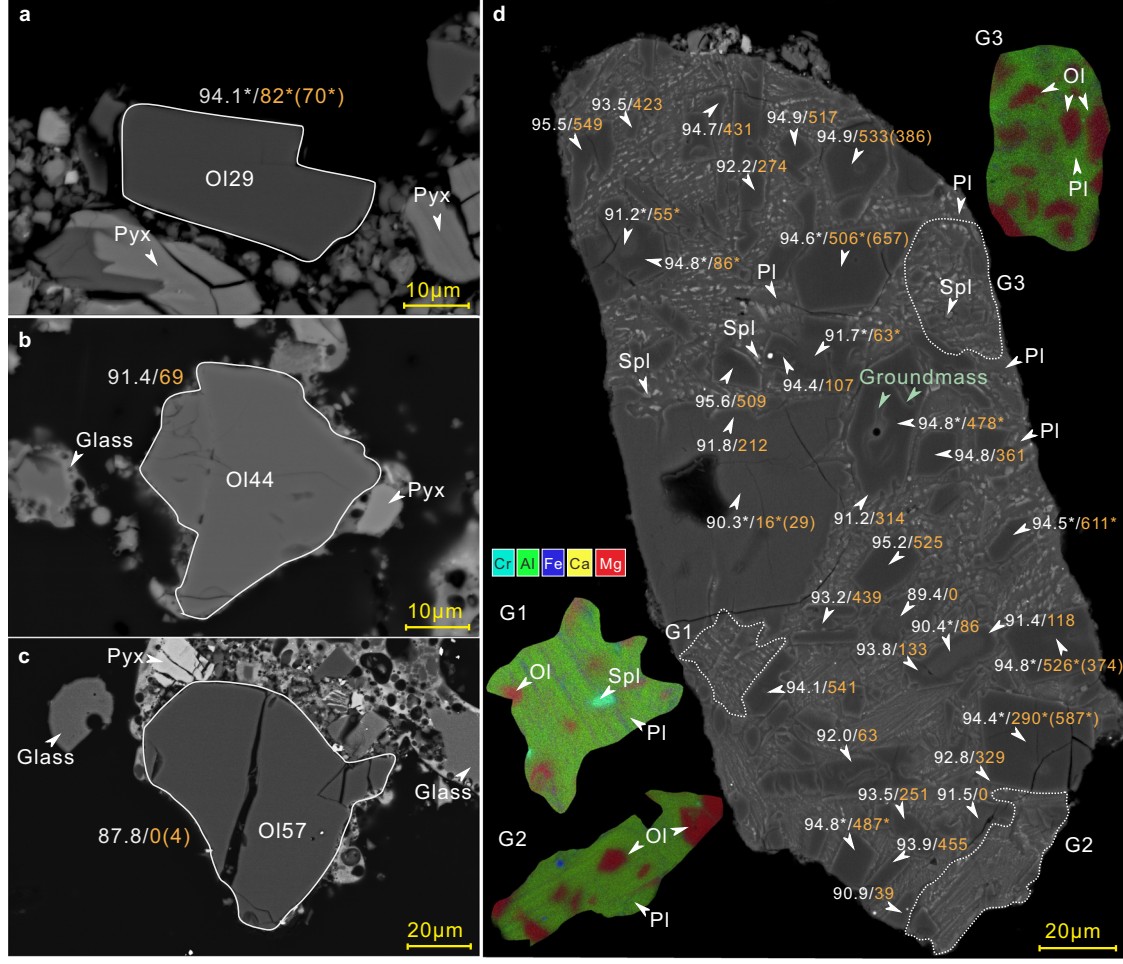

**Fig. 1 | Backscattered electron (BSE) images of the CE-6 olivines.**
**a**–**c** Representative olivine fragments and **d** olivine aggregates in the troctolitic groundmass. Numbers are the Fo values (white) and Ni contents (orange) of olivines (* stands for average values from multiple analyses on a single olivine grain). LA-ICPMS data are shown in parentheses if analyzed. G1-3 are representative energy

dispersive spectroscopy (EDS) mappings of the groundmass (outlined in dash lines). The inclusions in olivines have compositions align with the groundmass. Ol: olivine; Pl: plagioclase; Spl: spinel. Pyx, pyroxene. Data are reported in Supplementary Data 1.

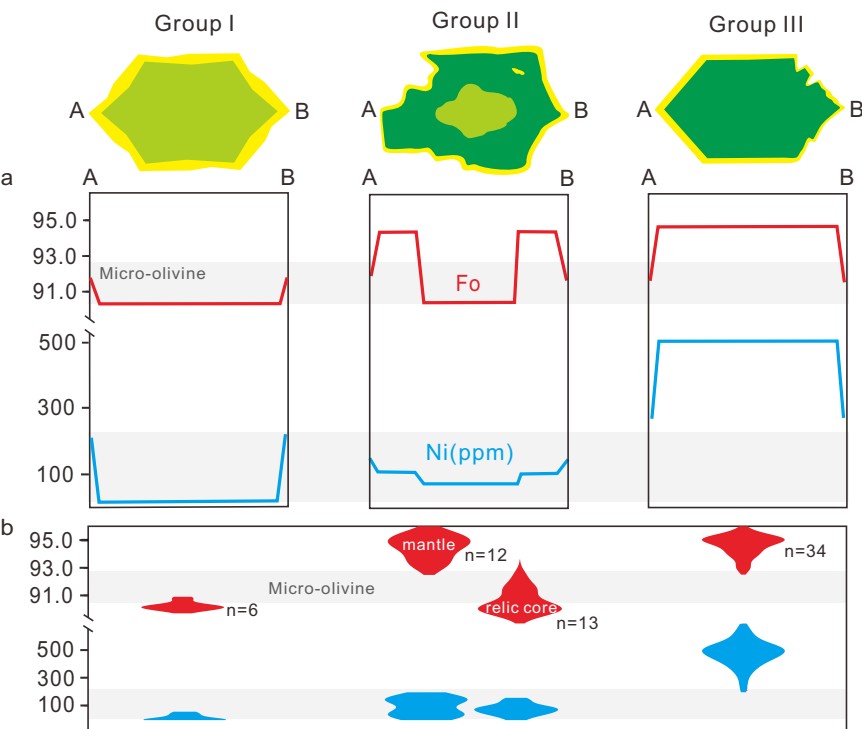

**Fig. 2 | Three populations of large olivine grains in the troctolitic groundmass. a** Schematic zoning profiles show the Fo and Ni patterns. Detail electron probe traverses across olivine crystals are reported in Supplementary Data 1 and shown in Supplementary Fig. 4. **b** Violin plots show the Fo and Ni of the olivine interior. The compositions of micro-olivine are shown as gray bars.

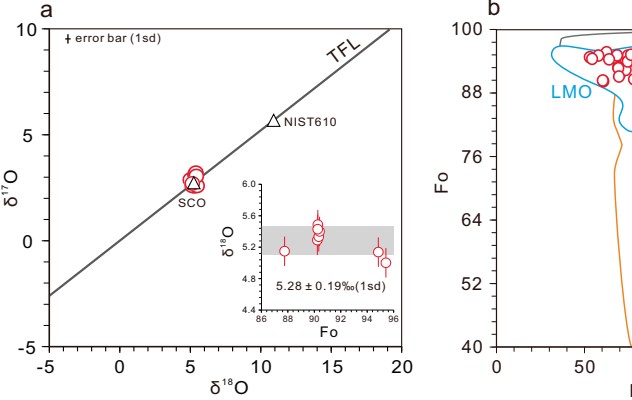

**Fig. 3 | Isotopic and elemental evidence supporting the lunar origin of the CE-6 olivines. a** Three oxygen isotopic compositions of the CE-6 olivines (circles). Also shown are the terrestrial fractionation line (TFL, black line), and oxygen isotopic compositions of the San Carlos olivine (SCO) and NIST 610 analyzed in this study (triangles). Error bars represent 1sd of SCO analyses. The inset in (**a**) is a plot of $\delta^{18}O$ versus Fo of the CE-6 olivines, with the gray bar representing the weighted average $\delta^{18}O$ value. **b** Fo versus FeO/MnO of the CE-6 olivines (circles). Compositional fields of lunar and ultramagnesian mafic fragment (UMMF) olivines are after ref. [35]. Compositional field of olivines from the Lunar Magma Ocean (LMO) experiments are from refs. [7–9,26]. The star represents the average olivine composition of the dunite in NWA 11421[2]. Data are reported in Supplementary Data 1 and 2.

hopper in shape, characterized by the highest Fo (92.9–95.6) and highest Ni (337–682 ppm) amongst all CE-6 olivines. Only a few olivine grains are large enough for precise measurements of Ni abundance using laser ablation–inductively coupled plasma mass spectrometry (LA-ICPMS). The EPMA data generally agree with the LA-ICPMS data (Supplementary Fig. 5 and Supplementary Data 1), i.e., bimodal distribution of Ni for olivine.

Oxygen isotopic compositions of the CE-6 olivines (mainly Group I and III), measured by using in-situ secondary ion mass spectrometry (SIMS), fall within the range defined by lunar rocks[32–34], plotting along the terrestrial fractionation line (TFL) with a homogenous $\delta^{18}O$ value of $5.28 \pm 0.19$ ‰ (1sd) (Fig. 3a and Supplementary Data 2).

## Indigenous and endogenous origin of the CE-6 olivines

Forsteritic olivines of exogenous origin have been reported in the lunar collection[35]. These olivines were characterized by high FeO/MnO (i.e., MnO below detection limit) and extremely low abundances of siderophile elements (e.g., <0.008 wt% Ni), which were interpreted as remnants of primitive chondritic meteorites impacted onto the lunar surface[35]. In contrast, the CE-6 olivines have a narrow FeO/MnO range

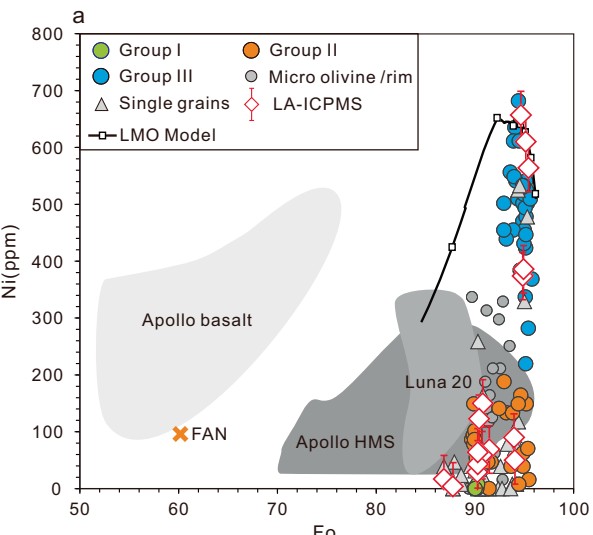
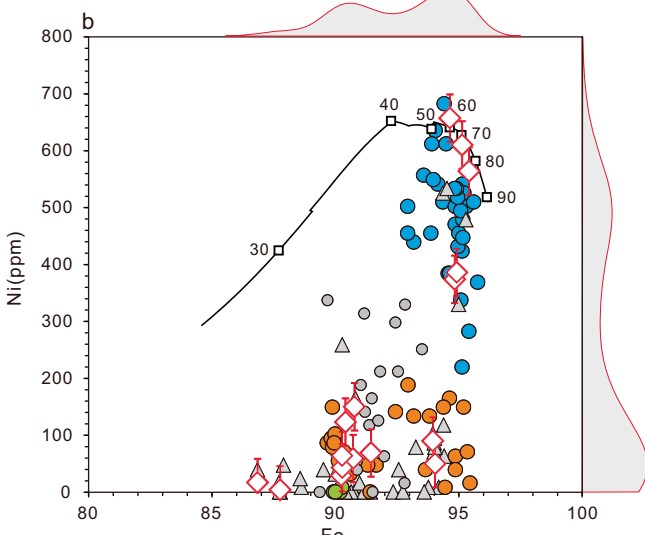

**Fig. 4 | Nickel contents versus Fo values of the CE-6 olivines. b** is a magnified view of **a** with fitted distribution curves of histograms showing the bimodal distribution of Fo and Ni in the CE-6 olivines. Olivines in lunar basalts[71], ferroan anorthosite[40] (FAN, cross) and high Mg olivines from the Luna 20[72] and Apollo Mg-suite rocks[40] (HMS) are also shown for comparison. EPMA analyses of olivine grains in the particular lithic fragment are shown in circles and those of single olivine grains are shown in triangles. LA-ICPMS data are shown in diamonds. Error bars represent 2se of LA-ICPMS analysis. The curve shows modeled composition of olivine crystallized from the LMO with an initial depth of 1000 km, with the squares representing the proportion of remaining LMO (%). Modeling details are described in Methods. Data are reported in Supplementary Data 1.

(53–123; Fig. 3b), consistent with those of other lunar olivines[35,36], and a wide range of Ni abundances (from zero to 682 ppm; Supplementary Data 1). The lack of mass-independent oxygen isotope fractionation in CE-6 olivines further supports their indigenous lunar origin. Fagan et al.[37] found that impact-derived olivines in Apollo 14 impact melts have higher Fo values compared to the pristine olivines from Apollo 14 and 12 high-alumina basalts. The lack of difference in Fo between olivine grains that coexist with the impact glass and those that do not (Supplementary Fig. 6), suggests that the high Fo of the CE-6 olivines is a pristine feature.

### Crustal and mantle origins of the CE-6 olivines

The most forsteritic pristine olivines in lunar crustal rocks are commonly associated with ultramafics and troctolites[38]. Their petrogenesis may be linked to the pressure-release melting of early LMO cumulates during mantle overturn, followed by crystallization at crustal depths[22,24,39,40]. A prominent feature of these forsteritic olivines in lunar crustal rocks is their Ni depletion compared to the less magnesian olivines in mare basalts[7,26,40] (Fig. 4). Longhi et al.[26] investigated the Ni abundance in Mg-suite olivines and found that the most forsteritic olivines in lunar ultramafics and troctolites have a maximum Ni content of approximately 300 ppm (Fig. 4). Several models have been proposed to explain the Ni depletion in Mg-suite olivines, such as remelting of LMO cumulate pile composed of primary dunite, norite and gabbronorite fluxed in part by KREEP (an acronym for the incompatibles K, rare-earth elements (REE), and P)[26] and small amounts of metal fractionation from the Mg-suite parental magmas prior to olivine crystallization[39]. Despite of different models, Ni depletion is considered to be a typical signature of olivines of Mg-suite origin.

The Group I and Group II olivines in our CE-6 collection have Ni contents below 300 ppm, falling within the range defined by Mg-suite olivines (Fig. 4). We tentatively assign these olivines to a Mg-suite origin. The Group II olivine exhibits a reverse zoning composed of a relic, Group I – like olivine core and a more magnesian mantle, suggesting a possible genetic relationship between the two groups. The reverse zoning can be formed by reaction of a pre-existing Group I -like olivine with a new pulse of hotter and more magnesian magma, such that the entrained olivines partially resorbed and become more Mg-

rich[41]. The Ni abundances of the mantles are only slightly elevated, indicating that the introduced magma is compositionally similar to the eruptive, Ni-depleted Mg-suite parent magma, but more magnesian. Overall, the Group I and Group II olivines may represent early formed Mg-suite olivine cumulates at the crustal level, which were later entrained into more Mg-rich eruptive lavas and transported to the surface. This formation mechanism, however, cannot explain the origin of the Group III olivine.

The Group III olivine, characterized by the highest Fo (Fo$_{92.9-95.6}$) and highest Ni contents (337–682 ppm), cannot be easily reconciled with typical Mg-suite origin. They may crystallize from an ultra-magnesian lava with Ni contents at least twice as high as those of typical Mg-suite parent magmas. However, no such ultra-magnesian lavas or basalts containing such high-Ni forsteritic olivines have been found in previous lunar collections[14,38]. Given our still limited understanding of lunar rock diversity, we cannot simply rule out the existence of such new type of ultra-magnesian and Ni-rich lavas. If present, the ultra-magnesian lava would have a similar genesis to the terrestrial komatiites[42], likely being produced by extensive melting of early mantle cumulates to achieve the most primitive signature of the Group III olivine (Supplementary Fig. 7). The hypothetical Ni-depletion event accounting for the origin of Mg-suite magmas would not have exerted an effect, in order to prevent the ultra-magnesian lava from Ni-depletion. Further in-depth study of the CE-6 returned samples and future lunar exploration would help test the possible existence of this new type of lava on the Moon. However, we propose below an alternative possibility for the origin of the Group III olivine: the first olivine crystallized from the LMO.

Laboratory experiments and numerical simulations of LMO solidification suggest that the Mg-rich olivine is the earliest crystallized mineral from the cooling magma ocean, with Fo varying from 96.0 to 87.6, corresponding to 0 to ~50% solidification[6–9,15–17]. Our simulation utilizes the Earth-like Lunar Primitive Upper Mantle (LPUM) composition[13] as the initial composition of the LMO. The Taylor Whole Moon (TWM) composition[43] has an apparently low Mg-number of 84 that is unable to produce olivine composition with Fo >94[7,9,15]. We also model the Ni content of olivine during the LMO solidification following a

well-established approach[26]. The empirical Beattie-Jones model[44,45] was employed to understand the partition of Ni between olivine and liquid: $D_{Ni}^{Ol/L} = 3.346 * D_{MgO}^{Ol/L} - 3.665$[44], in which the $D_{MgO}^{Ol/L} = MgO^{Ol}/MgO^L$ can be calculated at each step of LMO crystallization. Several studies estimated the initial Ni content of the LMO (also the Bulk Silicate Moon) to be depleted by a factor of about 3-4 compared to the terrestrial upper mantle, around $415 \pm 105$ ppm[14,46]. Consequently, our simulation on LMO crystallization is in accordance with previous attempts[7,26], demonstrating that the first crystallized olivine has a Fo value of 96.2 with Ni content of $509 \pm 131$ ppm (Fig. 4). The Ni content of olivine increases with ongoing LMO solidification and reaches a maximum at Fo of -92.4, after which the Ni content of olivine drops with decreasing Fo (Fig. 4). Compositions of the Group III olivines correspond well with the modeling results and thus could be of mantle origin. This inference is also supported by our additional modeling on Ni/Co evolution of olivines during LMO differentiation (Supplementary Fig. 8).

We acknowledge that the above two hypotheses for the origin of Group III olivine are not mutually exclusive. Given the equilibrium crystallization of olivine at the early stage of LMO solidification and the formation of the ultra-magnesian magma through high-degree partial melting of those olivine cumulates, the olivine crystallized from the ultra-magnesian magma is expected to have similar compositions to its source component, i.e., the early LMO cumulate (Supplementary Fig. 7). That is, the Group III olivine compositionally resembles the first olivine crystallized from the LMO.

### The travel of CE-6 olivines to the surface

Olivines are generally rare in the CE-6 soils (<0.5%)[21]. Orbital data of the SPA basin suggest that the upper mantle in that area is likely dominated by low-Ca pyroxene[47-49]. A possible explanation for the exposure of mantle olivines at the SPA basin would be that these olivines were entrained into the Mg-suite erupted lavas and brought to the surface (Fig. 5). At the onset of gravitational instability in the early history of the Moon, the less dense olivine cumulates of the LMO ascended and invaded into the upper mantle and experienced decompression melting to produce Mg-suite primary melts[22,24,39,40]. These Mg-suite primary melts are generally denser than the anorthositic crust and are thus predominantly intrusive, forming deep-seated Mg-suite intrusions[50]. The Group I olivine and the relic core of Group II olivine formed at this stage. Subsequent SPA basin-forming impact should have removed the upper anorthositic crust and excavated deep-seated mafic lithologies, making the major mafic compositional anomaly associated with the vast SPA basin[51,52]. The GRAIL data show that the crustal density at the SPA basin is higher than the Mg-suite primary melts[27,50], making it possible for the eruption of late ultramafic lavas. These high-degree, more magnesian eruptive lavas would have entrained mantle olivines (Group III) and crustal olivines (Group I), resulting in a mixed cargo of olivines formed at variable depths and conveyed them to the surface. Reaction of this Mg-rich magma with the less magnesian crustal olivines formed the Group II olivine. Future searching for potential exposure of mantle materials on the Moon should be focused on regions where Mg-suite extrusives are developed.

## Methods

### Tescan integrated mineral analysis

Major elemental mappings of the CE-6 soils were determined by TIMA at the Key Laboratory of Orogenic Belts and Crustal Evolution, School of Earth and Space Sciences, Peking University. The TIMA system comprises of a Tescan Mira Schottky field-emission scanning electron microscope with four silicon-drift energy-dispersive (EDS) detectors arranged at 90° intervals around the chamber. The measurements were performed in the high-resolution liberation analysis mode, and the backscattered electron image was obtained to identify individual particles and boundaries between distinct preliminary phases. A

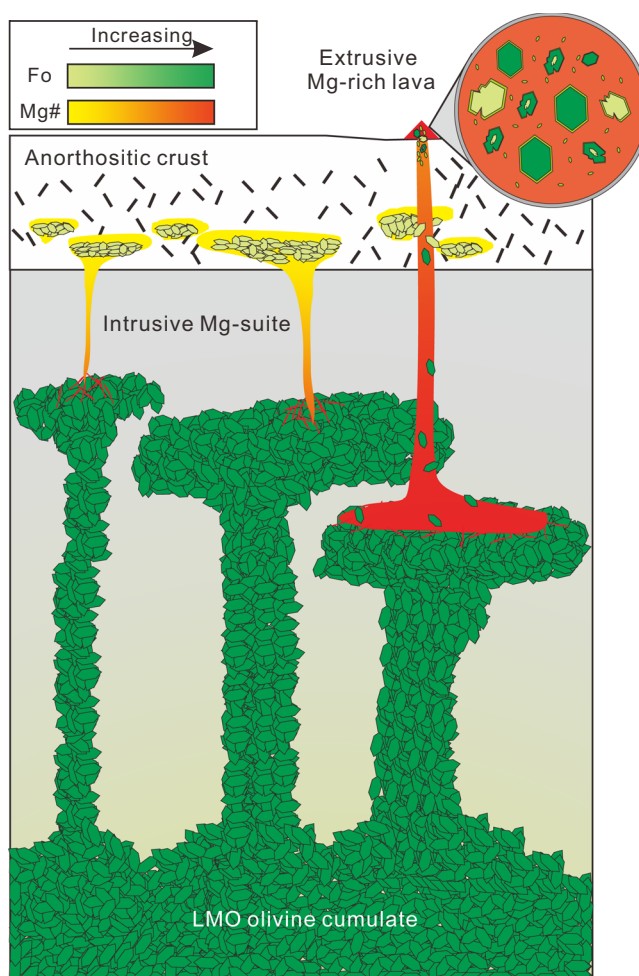

**Fig. 5 | Cartoon showing the transportation and exposure of mantle olivines in the SPA basin.** LMO olivine cumulates are shown in green whereas Mg-suite olivine cumulates are shown in yellow. See text for details.

rectangular mesh of measurements on each distinct phase was obtained with X-ray spectra. TIMA was performed at 25 kV using a spot size of -1 μm, a working distance of 15 mm, and a field size set at 1500 μm.

### Scanning electron microscopy and energy dispersive spectroscopy analyses

The fine backscattered electron (BSE) images and energy dispersive spectroscopy (EDS) mapping with semi-quantitative elemental analyses were performed using a Zeiss Supra 55 field-emission scanning electron microscope (FESEM) with an X-ray energy dispersive spectrometer, at the State Key Laboratory of Biogeology and Environmental Geology, China University of Geosciences, Beijing (CUGB). The EDS mappings with semi-quantitative elemental analyses were performed at an accelerating voltage of 20 kV and a working distance of 15 mm. Minerals as well as synthetic phases (MINM25-53) were used as standards. Data are reported in Supplementary Data 1.

### Electron microprobe analysis

The major and trace element compositions of olivines were determined using a JOEL JXA-iHP200F field emission EPMA electron microprobe at the Institute of Mineral Resources, Chinese Academy of Geological Sciences (CAGS). The olivines were analyzed at an acceleration voltage of 15 kV a probe current of 20 nA, and a focused beam diameter of -1 μm. The peak counting time was 100 s for Ni, 60 s for Ca, Al, Cr, Mn, Co and 10 s for Si, Mg, Fe, Ti, Na, K. The background time

was half of peak time. The detection limit was 0.006 wt.% for Ni, 0.028 wt.% for Co, 0.031 wt.% for Cr and 0.006–0.03 wt.% for other elements. A random set of olivine grains covering the entire range of Fo was further analyzed by using EPMA1720 electron microprobe at the EPMA Lab, CUGB. Both labs in CAGS and CUGB yield consistent data, further confirming the data quality (Supplementary Fig. 9). Data are reported in Supplementary Data 1.

## In situ trace element analysis

Siderophile elements including Ni, Co and Cr and Mn of olivines were further determined using laser ablation–inductively coupled plasma–mass spectrometry (LA-ICPMS) employing an Element XR HR-ICP-MS instrument coupled to a GeolasHD 193 nm ArF excimer LA system at the Institute of Geology and Geophysics, Chinese Academy of Sciences (IGGCAS). Helium was used as the ablation gas to improve the transport efficiency of the ablated aerosols. All measurements were conducted at 2 Hz frequency. The laser energy density was approximately 4.0 J cm⁻². The analyses were conducted using a laser diameter of 10 μm. ARM-1[53] reference glass was used for external calibration (Si was used as a standard for data reduction). Olivine MongOLSh11-2[54] and silicate glass GOR132-G[55] were used for quality control monitoring. Each set of standard samples, including ARM-1, GOR132-G and MongOLSh11-2, was analyzed once every 2 unknown samples were interspersed. The data obtained during ablation runs were processed using the Iolite 3.4 software with an in-house-built data reduction scheme mode[56] with the bulk normalization as 100 wt.% strategy. Data are reported in Supplementary Data 1. The analyses of reference materials MongOLSh11-2 and GOR132-G are given in Supplementary Data 3 and Supplementary Fig. 10, agreeing with published data. Their quoted uncertainties are defined as two times the standard error (2SE).

## In situ SIMS oxygen isotope analysis

Oxygen isotopic ratios of forsteritic olivines were analyzed using a CAMECA IMS-1280 multicollector ion probe at the IGGCAS. The Cs+ primary beam was accelerated at 10 kV with an intensity of ~2 nA. The spot is approximately 15 μm in diameter. An electron gun was used to compensate for sample charging during the analysis. Secondary ions were extracted at a potential of -10kV. Oxygen isotopes were measured in multi-collector mode with two off-axis Faraday cups. Each analysis consisted of 200 cycles with 2-s counting time. We analyzed the reference materials San Carlos olivine and NIST SRM 610 after analyzing every three unknown samples in the experiment to monitor analytical precision and calibrate instrumental mass fractionation. The average $\delta^{18}O$ value for San Carlos olivine is 5.25 ‰ in this study, which is similar to the reported certified value of 5.3 ‰[57]. The external reproducibility of the reference materials was 0.11–0.19 ‰ and 0.16–0.22 ‰ for $\delta^{18}O$ and $\delta^{17}O$ (1sd) over two days of analyses, respectively. Data are reported in Supplementary Data 2.

## Modeling of Ni and Co content of olivines during LMO solidification

We modeled the solidification of the LMO following a two-stage crystallization strategy[6,7,15,17,58]. The first 50 vol.% solidification process was assumed to be the equilibrium crystallization, followed by the fractional crystallization at >50 vol.% in steps with each representing 1/10th of the total magma ocean volume[58]. The initial depth of the LMO was assumed to be 1000 km based on geophysical constraints[11,59]. For every step, input pressure of the calculation was set to correspond to the middle depth of the LMO[58]. The oxygen fugacity was set at the iron-wüstite (IW) buffer. The first 50 vol.% solidification process was modeled by GeoPS[60] using the thermodynamic database HP633[61] and solution models for melt, olivine, orthopyroxene, clinopyroxene and feldspar from ref. 62. Calculated step was set as 300. The following fractional crystallization was modeled by the menu-driven alpha-MELTS interface (version 2.1) to run the subroutine version of rhyolite-

MELTS[63,64]. Phase equilibrium calculations from liquidus to solidus in 1 °C increments were performed.

For Ni and Co, the concentration of the early LMO during equilibrium crystallization at step m, $C_m^l$, is given by:

$$C_m^l = \frac{C_0^l * W_0}{W_m^l + D_{Ol}*W_m^{Ol} + D_{Opx}*W_m^{Opx}} \quad (1)$$

Where $C_0^l$ and $W_0$ stands for initial composition and mass of the LMO, respectively;

while the Ni and Co concentration of the early LMO during fractional crystallization is given by:

$$C_m^l = \frac{C_{m-1}^l * W_{m-1}^l}{W_m^l + D_{Ol}*W_m^{Ol} + D_{Opx}*W_m^{Opx}} \quad (2)$$

Where $C_{m-1}^l$ and $W_{m-1}^l$ stands for composition and remaining mass of the LMO at step m −1, respectively; $D_i$ and $W_i$ stands for partition coefficient and mass of phase i respectively. The initial Ni and Co content of the LMO were taken from ref. 46. Previous studies have revealed that the olivine-melt and orthopyroxene-melt partition coefficients of Ni and Co should be functions of $D_{Mg}$[44,45]. Thus, those partition coefficients would be functions of per cent solidified (PCS) of the LMO.

The Ni and Co contents of mantle olivines are then given by:

$$C_m^{Ol} = D_{Ol}*C_m^l \quad (3)$$

## Equilibrium melting modeling of the LMO cumulate pile at 50 per cent solidified

We chose the GeOPS to model the olivine composition during equilibrium melting model of the LMO cumulate pile at 50 PCS[7], from a LMO with an initial composition of the LPUM[13] (LPUMcp). Input pressure of the calculation was set to correspond to the depth of the remaining LMO at 50 PCS. The oxygen fugacity was set at the iron-wüstite (IW) buffer. The major elemental compositions of the LPUMcp were calculated from LMO crystallization modeling result in this study with 3% trapped instantaneous residual liquid (TIRL) at 50 PCS, and taken from ref. 7, respectively. The equilibrium melting was modeled using the thermodynamic database HP633[61] and solution models for melt, olivine, orthopyroxene, clinopyroxene and feldspar from ref. 62. Calculated step was set as 300.

The Ni concentration of the equilibrated melt during remelting of early LMO cumulates at step m, $C_m^l$, is given by:

$$C_m^l = \frac{C_s^0 * W_0}{W_m^l + D_{Ol}*W_m^{Ol} + D_{Opx}*W_m^{Opx}} \quad (4)$$

Where $C_s^0$ and $W_0$ stands for initial Ni content and mass of LPUMcp, respectively; $D_i$ and $W_i$ stands for partition coefficient and mass of phase i, respectively. The initial Ni content of the LPUMcp in this study was obtained by performing a Ni mass balance of the minerals and TIRL at 50 PCS. The initial Ni content of the LPUMcp from ref. 7 was obtained by first calculating the proportion of olivine, orthopyroxene and the remaining LMO liquid using the average mineral compositions at 50 PCS from their experiments and composition of the LPUMcp given in their study[7]. The Ni content of the remaining LMO liquid at 50 PCS can be calculated with initial Ni content of the LMO from ref. 46 through a mass balance calculation of the minerals and TIRL.

The Ni content of olivine equilibrated with the melt is then given by:

$$C_m^{Ol} = D_{Ol}*C_m^l \quad (5)$$

Model results are shown in Supplementary Fig. 7.

## Fractional crystallization modeling of the groundmass

PETROLOG program[65] was applied to perform crystal fractionation simulation on the groundmass. Our modeling was set as pure (100%) crystal fractionation at a constant pressure (1 bar). The calculation step was set as 1%. Mineral-melt equilibrium models used here are as follows: olivine[66], plagioclase[67], orthopyroxene[68], clinopyroxene[69] and spinel[70]. The $fo_2$ was set at the iron-wüstite (IW) buffer. The crystal fractionation modeling ceased when $Cr_2O_3 < 0$ wt.%. The crystallization trend of the groundmass in terms of Mg# of mafic minerals versus An of plagioclase is shown in Supplementary Fig. 2b; the crystallized mineral assemblages are consistent with petrological characteristics of the Mg-suite rocks and shown in Supplementary Fig. 2c.

## Data availability

All data generated in this study are provided in the Supplementary Information files.

## Code availability

The software and data files used for phase equilibria modeling are available at http://www.geops.org/zh-cn/ and https://magmasource.caltech.edu/forum/index.php/board,31.0.html. The software used for crystallization modeling can be accessed at http://petrolog.web.ru.

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

## Acknowledgements

We thank the China National Space Administration Agency for providing access to the Chang'e-6 returned sample (CE6C0400YJFM005). We are grateful to Dongjie Tang, Baozeng Xie, Chenming Wang and Xiaodan Chen for their support on EDS and EPMA analyses, and Guoqiang Tang for SIMS oxygen isotopic analysis. This project was jointly supported by the National Natural Science Foundation of China (42225301) to Q.L.L., and Fundamental Research Funds for the Central Universities (Grant 2652023001 and 2652023003) to S.J.W. B.Z. thanks the National nature Science Foundation of China (4224100313).

## Author contributions

S.J.W. conceptualized the study. S.Z.S. and J.X.H. collected the data. S.Z.S. performed geochemistry modeling. Q.L.L., H.W., S.T.W., Z.Y.C.,

J.H.H., B.Z., Y.S.H., and J.M.Z. contributed to the interpretation of the results. S.J.W. and S.Z.S. wrote the manuscript with input from all authors.

## Competing interests

The authors declare no competing interests.
