## [Transparent Peer Review file · Nature Communications]

Lunar primitive mantle olivine returned by Chang'e-6

Corresponding Author: Professor Shui-Jiong Wang

Version 0:

Reviewer comments:

Reviewer #1

(Remarks to the Author)

(1) Summary of strengths, weaknesses, and overall contribution

This study focuses on the identification and potential origin of forsteritic (Fo up to 95.6) olivine with high Ni abundance (374-657 ppm) in newly returned Chang'e-6 soils using quantitative (SEM, EPMA, LA-ICP-MS, SIMS) and qualitative (TIMA, SEM) analyses and geochemical modelling (pMELTS, PETROLOG). The identified Chang'e-6 olivine are lunar in origin as they fall along the $\delta^{18}\text{O}/\delta^{17}\text{O}$ terrestrial fractionation line (TFL). There are two populations of high Fo olivine with low Ni and high Ni abundance. High-Ni forsterite olivine are hypothesized to be fragments of early, lunar magma ocean (LMO) cumulates (i.e., fragments of the lunar mantle) entrained in a lunar Mg-suite like extrusive magma that brought the olivine to the surface within Chang'e-6 landing site. This hypothesis is supported by fractional crystallization and equilibrium geochemical modelling of the LMO at 20 kbar using lunar primitive upper mantle (LPUM) compositions. The origin for the low-Ni forsteritic olivine may be lunar Mg-suite in origin.

This research in this study is on a topic of relevance in the search for lunar mantle material, constraining the composition of the bulk-silicate Moon, and LMO crystallization. However, this paper has some major weaknesses that need to be addressed.

One major weakness is in the discussion of High-Ni versus Low-Ni forsterite olivine. It is unclear in Figure 1 which olivine phenocrysts or mineral fragments contain high-Ni abundance versus low-Ni abundance, which would change the interpretation of the results of this work. For example, in Figure 1D, some of the olivine show inverse zoning of rounded, relict olivine cores (Mg#=90) to more magnesian mantles (Mg#=93) to thin, bright outer rims that are more than likely enriched in Fe. Furthermore, some olivine in Figure 1D show embayment and dissolution in the cores of the minerals. Both inverse zoning and dissolution are evidence for early processes that would have altered the chemistry and history of the olivine minerals. The remaining olivine in Figure 1D that show only high Fo cores (Mg# > 93) could be the result of cutting effects in the thin section and are not true cores. Looking at Extended Data Table 1, olivine-OI01 has a core with average Mg# 90.1 and average low-Ni abundance of 16ppm but is mantled by Mg# 91 and Ni abundance 212. If the remaining high Fo, high-Ni olivine datapoints are of the mantle regions of the olivine, this would be indicative that the origin for this particular chemical signature is not early-LMO cumulates and more than likely the result of a secondary magma or geologic process. Additionally, if one focused on the olivine with relict, earlier formed cores that have Mg#s ~90 and low-Ni abundances, they are more in line with the Mg-Suite olivines discussed in this work. However, this paper does not address or discuss the inverse zoned olivine.

A second major weakness is not considering or discussing an impact origin for the olivine single mineral fragments that are often observed associated with glass in the BSE images (Figure 1, Extended Data Figure 1). Fagan et al. (2014) (DOI: <https://doi.org/10.1016/j.gca.2012.12.032>) discusses how different olivine chemical signatures can be used to discern impact-derived olivine versus endogenous. Typically, remelting of olivine would increase the Mg#, as observed in lunar impact melts. Additionally, the Chang'e-6 soils were collected from a mare basalt region of ~2.8 Ga in age that are similar to the mare basaltic meteorites with 2.8 Ga ages (Elardo et al., 2014; DOI: <https://doi.org/10.1111/maps.12239>) and have the current, highest Ni abundances when compared to Apollo basalts and Feldspathic samples. There may be a correlation

between the two and should be discussed further.

A third major weakness is the geochemical modelling used a definitive support for early LMO cumulate origin for the High Fo, High-Ni olivines. In geochemical modelling, there are multiple degrees of freedom that are not highlighted in both the discussion of the results and the methods. Additionally, for LMO modelling, the fractional crystallization at 20 kbar matched best with the results. However, LMO solidification may have had two-steps of crystallization, early equilibrium crystallization followed by fractional (Elkins-Tanton et al., 2011; DOI: <https://doi.org/10.1016/j.epsl.2011.02.004>). If the olivine identified in this work are products of early-LMO crystallization (0-50%), then the geochemical modelling should better reflect equilibrium crystallization as a potential crystallization pathway of the high-Fo olivine, but in Figure 3 and Extended Data Figure 4, this is not well represented.

I recommend the authors address the above major weaknesses before publication, especially major weakness 1, as this would change the interpretation of the identified olivine from lunar mantle origin to lunar Mg-Suite origin.

(2) Major Comments:

- Line 84: Add the Olivine ID's for the olivine in a-c similar to the Extended Data Fig. 1
- Line 93: It needs to be explained further why the UMMFs, with Olivines (Fo up to 98) are too high to be products of the LMO but not the olivines from this study (Fo up to 96).
- Line 114: This paragraph introduces the different lunar rock suites that have primitive olivine (Fo >90) but does not discuss in detail the high forsteritic olivine in the lunar Mg-suite whose olivine overlap in Mg# with the olivine of this work. This should be introduced in this paragraph.
- Line 142-144: Some low-Ti mare basalts, in-particular, the low-Ti mare basaltic meteorites that have high Ni abundances, may be the product of late-stage LMO cumulates (85-96% LMO Crystallization) and thus, the comparison made in this sentence is misleading. Elaborating which low-Ti mare basalts are being extrapolated for Ni abundances needs to be discussed here.
- Line 217: This is a little confusing since it is not clearly distinguished in the BSE images which olivine are low-Ni versus high-Ni. Therefore, the readers cannot compare texture.
- Line 301: References are missing here to refer to which previous studies used 20 kbar

(3) Minor Comments:

- Line 36: Lunar mantle is a fundamental part in the puzzle of Moons formation and differentiation.
 - o Insert "Constraining the lunar mantle is fundamental to understanding the Moon's formation, differentiation, and evolution."
 - o Reference used is Moriarty III et al. (2021) which focuses on the search for the lunar mantle in remote sensing.
- Line 38: insert "the" in front of "Apollo-mission samples"
- Line 38: Change "Unfortunately" to "However"
- Line 38: Change "definitively" to "definitive"
- Line 40: Our understanding of the lunar mantle is also based on coordinated analyses of volcanic products (i.e., mare basalts, volcanic glass beads) whom partial melts are products of mantle cumulates. This sentence needs to be restructured to reflect ways the lunar mantle is studied.
- Line 40: Change "Physico-chemical" to "Physio-chemical"
- Line 41: Remove "however"
- Line 42: Add "are" before "subject to change"
- Line 44: "Classic lunar magma ocean (LMO) concept" can be changed to "The Lunar Magma Ocean (LMO) paradigm"
- Line 45: "Magnesian olivine-dominated lower mantle" are mentioned here but then rephrased as "dunite cumulates" in line 47. To reflect consistency, either add the type of magnesian olivine lithologies predicted in by models, such as: (i.e., dunite cumulates, ect.) or change dunite cumulates in line 47 to less dense mafic cumulates.
- Line 48: Shallow mantle here might not be the right phrase. Consider rephrasing to include density instability alongside gravitational restructuring.
- Line 49: This sentence needs references
- Line 49: There isn't a consensus to what mineral assemblages are in the lunar upper mantle but examples could be provided, for example: Elkins-Tanton et al., 2011 discusses and compares the mineral assemblages for the lunar mantle in different LMO models.
- Line 50: Rephrase to reflect that potential excavated upper lunar mantle material has been identified in the SPA basin. References: Moriarty III et al., 2021
- Line 76: Remove ", and" after "textures, and"
- Line 93: "Olivines with Fo high" should be "Olivines with high Fo"
- Line 128: Change "Now that the question is" to "It is unclear"
- Line 128: Remove ",," from "LMO cumulates,"
- Line 130: Remove "We will tackle this issue by using Ni content in olivine"
- Line 135: Need references for the lunar DNi diffusion coefficients
- Line 152: Figure 3: The "Apollo Mg-Suite" Field is missing "Mg-Suite"
- Line 165-167: This isn't the only hypothesis for the formation of the Mg-suite troctolites. Rephrasing "whose petrogenesis is linked" to "whose petrogenesis may be linked" would better fit this sentence.
- Line 168: Remove ",," from "Mg-suite olivines,"
- Line 172: Remove ",," from "by KREEP55,"
- Line 173: Change "Despite the debate on the Ni depletion problem" to "Despite competing hypotheses on Ni depletion in Mg-suite olivines,"
- Line 196: Remove ",," from "upper mantle,"
- Line 209: Change "unambiguous" to "ambiguous"

- Line 215: Remove “,” from “primitive melt,”

Reviewer #2

(Remarks to the Author)
See attached review file.

[Editorial Note: This attachment is displayed on the final three pages of this file]

Version 1:

Reviewer comments:

Reviewer #1

(Remarks to the Author)
Thank you to the authors for addressing all comments. I especially appreciate the additional analyses, figures, and modelling to further justify their argument.

I recommend this manuscript for publication.

Response to comments

REVIEWER COMMENTS

Reviewer #1 (Remarks to the Author):

(1) Summary of strengths, weaknesses, and overall contribution

This study focuses on the identification and potential origin of forsteritic (Fo up to 95.6) olivine with high Ni abundance (374-657 ppm) in newly returned Chang'e-6 soils using quantitative (SEM, EPMA, LA-ICP-MS, SIMS) and qualitative (TIMA, SEM) analyses and geochemical modelling (pMELTS, PETROLOG). The identified Chang'e-6 olivine are lunar in origin as they fall along the $\delta^{18}\text{O}/\delta^{17}\text{O}$ terrestrial fractionation line (TFL). There are two populations of high Fo olivine with low Ni and high Ni abundance. High-Ni forsterite olivine are hypothesized to be fragments of early, lunar magma ocean (LMO) cumulates (i.e., fragments of the lunar mantle) entrained in a lunar Mg-suite like extrusive magma that brought the olivine to the surface within Chang'e-6 landing site. This hypothesis is supported by fractional crystallization and equilibrium geochemical modelling of the LMO at 20 kbar using lunar primitive upper mantle (LPUM) compositions. The origin for the low-Ni forsteritic olivine may be lunar Mg-suite in origin.

This research in this study is on a topic of relevance in the search for lunar mantle material, constraining the composition of the bulk-silicate Moon, and LMO crystallization. However, this paper as some major weaknesses that needs to be addressed.

One major weakness is in the discussion of High-Ni versus Low-Ni forsterite olivine. It is unclear in Figure 1 which olivine phenocrysts or mineral fragments contain high-Ni abundance versus low-Ni abundance, which would change the interpretation of the results of this work. For example, in Figure 1D, some of the olivine show inverse zoning of rounded, relict olivine cores (Mg#=90) to more magnesian mantles (Mg#=93) to thin, bright outer rims that are more than likely enriched in Fe. Furthermore, some olivine Figure 1D show embayment and dissolution in the cores of the minerals. Both inverse zoning and dissolution are evidence for early processes that would have altered the chemistry and history of the olivine minerals. The remaining olivine in Figure 1D that show only high Fo cores (Mg# > 93) could be the result of cutting effects in the thin section and are not true cores. Looking at Extended Data Table 1, olivine- O101 has a core with average Mg# 90.1 and average low-Ni abundance of 16 ppm but is mantled by Mg# 91 and Ni abundance 212. If the remaining high Fo, high-Ni olivine datapoints are of the mantle regions of the olivine, this would be indicative that the origin for this particular chemical signature is not early-LMO cumulates and more than likely the result of a secondary magma or geologic process. Additionally, if one focused on the olivine with relict, earlier formed cores that have Mg#s \approx 90 and low-Ni abundances, they are more in line with the Mg-Suite olivines discussed in this work. However, this paper does not address or discuss the inverse zoned olivine.

Reply: Thanks for this useful comment. We acknowledge that the olivines in the CE-6 soils are compositionally and texturally complicated. Following this suggestion, we did additional rim-core-rim EPMA analyses of olivines, hoping to have a better classification and understanding of the olivine populations in the CE-6 soils. Based on morphological and geochemical characteristics, we categorized the large olivine grains in the lithic fragment into three types (Fig.1 below): **“Group I olivine is represented by the largest polygonal crystal (~50 μm) with a dissolution feature at the margin. Its interior is homogenous, with relatively low Fo (89.9-90.8) and low Ni (0-55 ppm); Group II olivine is predominantly hopper-shaped, containing an irregular, relict core featured by low Fo (89.7-92.4) and low Ni (0-149 ppm), and a more magnesian mantle with higher Fo (92.9-95.4) but similar Ni contents (8-188 ppm); Group III olivine is polyhedral to hopper in shape, characterized by the highest Fo (92.9-95.6) and highest Ni (337-682 ppm) amongst all CE-6 olivines.”**

Fig. 1. Three populations of large olivine grains in the lithic fragment. (a) Schematic zoning profiles show the Fo and Ni patterns. **(b)** Violin plots show the Fo and Ni of the olivine interior. The compositions of micro-olivine are shown as grey bars. Detail electron probe traverses across olivine crystals are reported in the Supplementary Table 1 and shown in Supplementary Fig. 4.

The compositions of **Group I** and **Group II** olivines fall within the range defined by Mg-suite olivines, indicating a Mg-suite origin. The reverse zoning in **Group II** olivines can be formed by reaction of a pre-existing **Group I**-like olivine with a more magnesian magma. For details, please see our revision in Lines 165 to 177:

*“The **Group I** and **Group II** olivines in our CE-6 collection have Ni contents below 300 ppm, falling within the range defined by Mg-suite olivines (Fig. 4). We tentatively assign these olivines to a Mg-suite origin. The **Group II** olivine exhibits a reverse zoning composed of a relic, **Group I** – like olivine core and a more magnesian mantle, suggesting a possible genetic relationship between the two groups. The reverse zoning can be formed by reaction of a pre-existing **Group I**-like olivine with a new pulse of hotter and more magnesian magma, such that the entrained olivines partially resorbed and become more Mg-rich⁴¹. The Ni abundances of the mantles are only slightly elevated, indicating that the introduced magma is compositionally similar to the eruptive, Ni-depleted Mg-suite parent magma, but more magnesian. Overall, the **Group I** and **Group II** olivines may represent early formed Mg-suite olivine cumulates at the crustal level, which were later entrained into more Mg-rich eruptive lavas and transported to the surface.”*

However, **Group III** olivines, with the highest Fo and highest Ni abundance, have a different origin against **Group I** and **Group II** olivines. We propose that the compositions of **Group III** olivines resemble the first olivine crystallized from the Lunar Magma Ocean. For details, please see our revision in Lines 178 to 221:

*“The **Group III** olivine, characterized by the highest Fo (Fo92.9-95.6) and highest Ni contents (337-682 ppm), cannot be easily reconcealed with typical Mg-suite origin. They may crystallize from an ultra-magnesian lava with Ni contents at least twice as high as those of typical Mg-suite parent magmas. However, no such ultra-magnesian lavas or basalts containing such high-Ni forsteritic olivines have been found in previous lunar collections^{14,38}. Given our still limited understanding of lunar rock diversity, we cannot simply rule out the existence of such new type of ultra-magnesian and Ni-rich lavas. If present, the ultra-magnesian lava would have a similar genesis to the terrestrial komatiites⁴², likely being produced by extensive melting of early mantle cumulates to achieve the most primitive signature of the **Group III** olivine (Supplementary Fig. 7). The hypothetical Ni-depletion*

event accounting for the origin of Mg-suite magmas would not have exerted an effect, in order to prevent the ultra-magnesian lava from Ni-depletion. Further in-depth study of the CE-6 returned samples and future lunar exploration would help test the possible existence of this new type of lava on the Moon. However, we propose below an alternative possibility for the origin of the **Group III** olivine: the first olivine crystallized from the LMO... We acknowledge that the above two hypotheses for the origin of **Group III** olivine are not mutually exclusive. Given the equilibrium crystallization of olivine at the early stage of LMO solidification and the formation of the ultra-magnesian magma through high-degree partial melting of those olivine cumulates, the olivine crystallized from the ultra-magnesian magma is expected to have similar compositions to its source component, i.e., the early LMO cumulate (Supplementary Fig. 7). That is, the **Group III** olivine resembles the first olivine crystallized from the LMO in terms of composition.”

A second major weakness is not considering or discussing an impact origin for the olivine single mineral fragments that are often observed associated with glass in the BSE images (Figure 1, Extended Data Figure 1). Fagan et al. (2014) (DOI: <https://doi.org/10.1016/j.gca.2012.12.032>) discusses how different olivine chemical signatures can be used to discern impact-derived olivine versus endogenous. Typically, remelting of olivine would increase the Mg#, as observed in lunar impact melts. Additionally, the Chang-e’6 soils were collected from a mare basalt region of ~2.8 Ga in age that are similar to the mare basaltic meteorites with 2.8 Ga ages (Elardo et al., 2014; DOI: <https://doi.org/10.1111/maps.12239>) and have the current, highest Ni abundances when compared to Apollo basalts and Feldspathic samples. There may be a correlation between the two and should be discussed further.

Reply: Thanks for bringing us Fagan et al. paper. Fagan et al. (2013) found that olivines in Apollo 14 impact melts have higher Fo values compared to the pristine olivines from Apollo 14 and 12 high-alumina basalts. Nonetheless, the lack of difference in Fo between olivine grains that coexist with the impact glass and those that do not, suggests that the high Fo of the CE-6 olivines is a pristine feature (Fig. 2 below). Please see our revision in Lines 138 to 142:

“Fagan et al.³⁷ found that impact-derived olivines in Apollo 14 impact melts have higher Fo values compared to the pristine olivines from Apollo 14 and 12 high-alumina basalts. The lack of difference in Fo between olivine grains that coexist with the impact glass and those that do not (Supplementary Fig. 6), suggests that the high Fo of the CE-6 olivines is a pristine feature.”

Fig. 2 Fo versus Ni content of the single olivine fragments coexist with the impact glass (filled) compared with those do not (unfilled).

With respect to the basaltic meteorites with 2.8 Ga ages, we total agree that they may be potentially linked to the Chang'e 6 landing site. Although these 2.8 Ga basaltic meteorites have the highest Ni abundance among lunar basalts and meteorites, olivines therein have significantly lower Fo (34-65) comparing with the high Fo and high Ni olivines (92.9-95.6). Therefore, we consider that they have a different origin with the high Fo and high Ni olivines in this study.

A third major weakness is the geochemical modelling used a definitive support for early LMO cumulate origin for the High Fo, High-Ni olivines. In geochemical modelling, there are multiple degrees of freedom that are not highlighted in both the discussion of the results and the methods. Additionally, for LMO modelling, the fractional crystallization at 20 kbar matched best with the results. However, LMO solidification may have had two-steps of crystallization, early equilibrium crystallization followed by fractional (Elkins-Tanton et al., 2011; DOI: <https://doi.org/10.1016/j.epsl.2011.02.004>). If the olivine identified in this work are products of early-LMO crystallization (0-50%), then the geochemical modelling should better reflect equilibrium crystallization as a potential crystallization pathway of the high-Fo olivine, but in Figure 3 and Extended Data Figure 4, this is not well represented.

Reply: Thanks for this comment. Following your suggestion, we have revised the model to a two-stage LMO solidification: The first 50 vol.% solidification process is assumed to be the equilibrium crystallization, followed by the fractional crystallization at >50 vol.%. The initial depth of the LMO is assumed to be 1000km based on geophysical constraints (Zhao et al., 2012; Harada et al., 2014). The oxygen fugacity was set at the iron-wüstite (IW) buffer. Our revised modeling results align well with the high-Fo, high-Ni olivines, indicating that the compositions of **Group III** olivines resemble those equilibrated and crystallized from the early LMO (Fig. 3 below). For the modeling details please see the **Methods** in Lines 330 to 341:

“We modeled the solidification of the LMO following a two-stage crystallization strategy^{6,7,15,17,58}. The first 50 vol.% solidification process was assumed to be the equilibrium crystallization, followed by the fractional crystallization at >50 vol.% in steps with each representing 1/10th of the total magma ocean volume⁵⁸. The initial depth of the LMO is assumed to be 1000km based on geophysical constraints^{11,59}. For every step, input pressure of the calculation was set to correspond to the middle depth of the LMO⁵⁸. The oxygen fugacity was set at the iron-wüstite (IW) buffer. The first 50 vol.% solidification process was modeled by GeoPS⁶⁰ using the thermodynamic database HP633⁶¹ and solution models for melt, olivine, orthopyroxene, clinopyroxene and feldspar from ref. ⁶². Calculated step was set as 300. The following fractional crystallization was modeled by the menu-driven alphaMELTS interface (version 2.1) to run the subroutine version of rhyolite-MELTS^{63,64}. Phase equilibrium calculations from liquidus to solidus in 1°C increments were performed.”

Fig. 3 Nickel content versus Fo value of the CE-6 olivines. **(b)** magnified view of the panel **(a)** with a histogram showing the bimodal distribution of Fo and Ni in CE-6 olivines. Modeled compositions of olivine crystallized from the LMO with initial depth of 1000 km is shown in the solid line. The squares stand for the proportion of remaining LMO (%).

I recommend the authors address the above major weaknesses before publication, especially major weakness 1, as this would change the interpretation of the identified olivine from lunar mantle origin to lunar Mg-Suite origin.

Reply: We appreciate the valuable comments and suggestions, and have revised the manuscript accordingly.

(2) Major Comments:

- Line 84: Add the Olivine ID's for the olivine in a-c similar to the Extended Data Fig. 1.

Reply: Thanks. Done.

- Line 93: It needs to be explained further why the UMMFs, with Olivines (Fo up to 98) are too high to be products of the LMO but not the olivines from this study (Fo up to 96).

Reply: We have now revised the discussion of UMMFs. Please see Lines 130 to 137 for the revision:

“Forsteritic olivines of exogenous origin have been reported in the lunar collection³⁵. These olivines were characterized by high FeO/MnO (i.e., MnO below detection limit) and extremely low abundances of siderophile elements (e.g., <0.008 wt% Ni), which were interpreted as remnants of primitive chondritic meteorites impacted onto the lunar surface³⁵. In contrast, the CE-6 olivines have a narrow FeO/MnO range, consistent with those of other lunar olivines^{35,36} (53 to 123; Fig. 3b), and a wide range of Ni abundances (from zero to 682 ppm; Supplementary Table 1). The lack of mass-independent oxygen isotope fractionation in CE-6 olivines further supports their indigenous lunar origin.”

- Line 114: This paragraph introduces the different lunar rock suites that have primitive olivine (Fo >90) but does not discuss in detail the high forsteritic olivine in the lunar Mg-suite whose olivine overlap in Mg# with the olivine of this work. This should be introduced in this paragraph.

Reply: Thanks. We have now revised the discussion of the olivine with Mg-suite origin. Please see the revision in Lines 152 to 164:

“The most forsteritic pristine olivines in lunar crustal rocks are commonly associated with ultramafics

and troctolites³⁸. Their petrogenesis may be linked to the pressure-release melting of early LMO cumulates during mantle overturn, followed by crystallization at crustal depths^{22,24,39,40}. A prominent feature of these forsteritic olivines in lunar crustal rocks is their Ni depletion compared to the less magnesian olivines in mare basalts^{7,26,40} (Fig. 4). Longhi et al.²⁶ investigated the Ni abundance in Mg-suite olivines and found that the most forsteritic olivines in lunar ultramafics and troctolites have a maximum Ni content of approximately 300 ppm (Fig. 4). Several models have been proposed to explain the Ni depletion in Mg-suite olivines, such as remelting of LMO cumulate pile composed of primary dunite, norite and gabbronorite fluxed in part by KREEP²⁶ and small amounts of metal fractionation from the Mg-suite parental magmas prior to olivine crystallization³⁸. Despite of different models, Ni depletion is considered to be a typical signature of olivines of Mg-suite origin.”

• Line 142-144: Some low-Ti mare basalts, in-particular, the low-Ti mare basaltic meteorites that have high Ni abundances, may be the product of late-stage LMO cumulates (85-96% LMO Crystallization) and thus, the comparison made in this sentence is misleading. Elaborating which low-Ti mare basalts are being extrapolated for Ni abundances needs to be discussed here.

Reply: Thanks. We have now focused on the discussion on the origin of the three groups of olivine in the revision, and removed the low-Ti mare basalt part.

• Line 217: This is a little confusing since it is not clearly distinguished in the BSE images which olivine are low-Ni versus high-Ni. Therefore, the readers cannot compare texture.

Reply: Thanks for this comment. We have now added the Ni content of the olivines in Fig. 1 (**Fig. 4 below**) as well as in Supplementary Fig. 1. We also made new rim-core-rim EPMA analyses on the olivine grains and show in Fig. 2 in the main text (**Fig. 1 above**) and Supplementary Fig. 4.

Fig. 4. Backscattered electron (BSE) images of representative olivine fragments (a-c) and olivine aggregates in the troctolitic groundmass (d). Numbers are the Fo values (white) and Ni contents (orange) of olivines (* stands for average values from multiple analyses on a single olivine grain). LA-ICPMS data were shown in parentheses if analyzed. G1-3 are representative energy dispersive spectroscopy (EDS) mappings of the groundmass (outline in dash lines). The inclusions in olivines have compositions align with the groundmass.

- Line 301: References are missing here to refer to which previous studies used 20 kbar.

Reply: We have now modeled LMO crystallization assuming an initial LMO depth of 1000km based on the geophysical constraints (Zhao et al., 2012; Harada et al., 2014). The input pressure was set to correspond to the middle depth of the LMO (Arai et al., 2017).

(3) Minor Comments:

- Line 36: Lunar mantle is a fundamental part in the puzzle of Moons formation and differentiation.
 - o Insert “Constraining the lunar mantle is fundamental to understanding the Moon’s formation, differentiation, and evolution.”
 - o Reference used is Moriarty III et al. (2021) which focuses on the search for the lunar mantle in remote sensing.

Reply: Done.

- Line 38: insert “the” in front of “Apollo-mission samples”

- Line 38: Change “Unfortunately” to “However”
- Line 38: Change “definitively” to “definitive”

Reply: Done.

- Line 40: Our understanding of the lunar mantle is also based on coordinated analyses of volcanic products (i.e., mare basalts, volcanic glass beads) whom partial melts are products of mantle cumulates. This sentence needs to be restructured to reflect ways the lunar mantle is studied.
- Line 40: Change “Physico-chemical” to “Physio-chemical”

Reply: Thanks. Done. Please see our revision in Lines 47 to 49:

“Our understanding of the ancient and modern lunar mantle is largely based on the physio-chemical models, laboratory experiments, geophysical and remote sensing observations, and coordinated analyses of volcanic products⁵⁻¹⁷.”

- Line 41: Remove “however”

Reply: Done.

- Line 42: Add “are” before “subject to change”

Reply: Done.

- Line 44: “Classic lunar magma ocean (LMO) concept” can be changed to “The Lunar Magma Ocean (LMO) paradigm”

Reply: Done.

- Line 45: “Magnesian olivine-dominated lower mantle” are mentioned here but then rephrased as “dunite cumulates” in line 47. To reflect consistency, either add the type of magnesian olivine lithologies predicted in by models, such as: (i.e., dunite cumulates, ect.) or change dunite cumulates in line 47 to less dense mafic cumulates.

Reply: Done.

- Line 48: Shallow mantle here might not be the right phrase. Consider rephrasing to include density instability alongside gravitational restructuring.

Reply: Thanks. Done. Please see our revision in Lines 54 to 56:

“However, as the Moon evolved, this initial cumulate stratigraphy was disrupted: the less dense olivine cumulates would have ascended upon the overlying high-density mantle via gravitational restructuring¹⁹.”

- Line 49: This sentence needs references
- Line 49: There isn’t a consensus to what mineral assemblages are in the lunar upper mantle but

examples could be provided, for example: Elkins-Tanton et al., 2011 discusses and compares the mineral assemblages for the lunar mantle in different LMO models.

Reply: Thanks. We have now added several references focusing on the mineral assemblages for the lunar mantle. Please see the revision in Lines 56 to 58:

“There is currently no consensus on what lithologies of the lunar upper mantle would have been^{1,5,20}.”

- Line 50: Rephrase to reflect that potential excavated upper lunar mantle material has been identified in the SPA basin. References: Moriarty III et al., 2021

Reply: Thanks. Please see our revision in Lines 59 to 60:

“The South Pole-Aitken Basin (SPA) on the Moon’s farside is believed to be the most promising location for mantle materials to be exposed on the lunar surface¹.”

- Line 76: Remove “, and” after “textures, and”

Reply: Done.

- Line 93: “Olivines with Fo high” should be “Olivines with high Fo”

Reply: We have removed this sentence.

- Line 128: Change “Now that the question is” to “It is unclear”

- Line 128: Remove “,” from “LMO cumulates,”

Reply: We have removed this sentence.

- Line 130: Remove “We will tackle this issue by using Ni content in olivine”

Reply: Done.

- Line 135: Need references for the lunar DN_{Ni} diffusion coefficients

Reply: Thanks. Done. Please see the revision in Lines 201 to 204:

*“The empirical Beattie-Jones model^{44,45} was employed to understand the partition of Ni between olivine and liquid: $D_{Ni}^{Ol/L} = 3.346 * D_{MgO}^{Ol/L} - 3.665^{44}$, in which the $D_{MgO}^{Ol/L} = MgO^{Ol}/MgO^L$ can be calculated at each step of LMO crystallization.”*

- Line 152: Figure 3: The “Apollo Mg-Suite” Field is missing “Mg-Suite”

Reply: Thanks, Done.

- Line 165-167: This isn’t the only hypothesis for the formation of the Mg-suite troctolites. Rephrasing “whose petrogenesis is linked” to “whose petrogenesis may be linked” would better fit this sentence.

Reply: Thanks. Done.

- Line 168: Remove “,” from “Mg-suite olivines,”

Reply: Done.

- Line 172: Remove “,” from “by KREEP55,”

Reply: Done.

- Line 173: Change “Despite the debate on the Ni depletion problem” to “Despite competing hypotheses on Ni depletion in Mg-suite olivines,”

Reply: Thanks. Please see Lines 163 to 164 for the revision:

“Despite of different models, Ni depletion is considered to be a typical signature of olivines of Mg-suite origin.”

- Line 196: Remove “,” from “upper mantle,”

Reply: Done.

- Line 209: Change “unambiguous” to “ambiguous”

Reply: We have removed this sentence.

- Line 215: Remove “,” from “primitive melt,”

Reply: We have removed this sentence.

Reviewer #2 (Remarks to the Author):

Review of Nature Communications submission 559489: “Lunar primitive mantle olivine returned by Chang’e-6” by Si-Zhang Sheng et al.

I recommend that the manuscript would be acceptable for publication, after revisions. I’ve attached a version of the submitted manuscript with a few edits to the text, a few additional comments, and some suggestions of additional references.

Reply: We appreciate the valuable comments and suggestions provided by the reviewer.

This manuscript reports intriguing high-Mg olivine crystals from the regolith returned by the Chang’e-6 spacecraft from the Moon’s South-Pole Aitkin Basin. The data are an important contribution to understanding the Moon, but I am not convinced that the high-Mg olivines are necessarily samples of the Moon’s mantle. The authors come to this inference because the olivine grains are very magnesian, $Mg^* = \text{molar Mg}/(\text{Mg}+\text{Fe})$ up to almost 96%, and have high Ni abundances, up to ~650 ppm. No other

known lunar olivines have both these characteristics, neither basalts nor basaltic plutonic rocks. The authors show that such olivine compositions could have formed in cumulate rocks from the early Lunar Magma Ocean.

This argument is reasonable, and I have no serious problems with it. However, there is another possible interpretation that ought to be mentioned in the manuscript; an interpretation which (sadly) changes the focus and apparent importance of the olivines. It is reasonable that the magnesian nickeloan olivines crystallized from ultrabasic lavas unlike those discovered so far. This might seem far-fetched, but no more so than finding fragments of mantle olivine.

Reply: Thank you for this comment. We acknowledge that the high Fo, high Ni olivines could have crystallized from an unrecognized ultra-magnesian magma. We have made additional rim-core-rim analyses of the large olivines in the lithic fragment, in order to have a better understanding of the olivine populations and their origins. We classified the olivines in the lithic fragment into three groups (please see Fig. 1 above), and attributed the **Group I** and **Group II** olivines to a Mg-suite origin, whereas the **Group III** olivine has a different origin. We have expanded the discussion in this revision accordingly. Please see Lines 178 to 221:

*“The **Group III** olivine, characterized by the highest Fo ($Fo_{92.9-95.6}$) and highest Ni contents (337-682 ppm), cannot be easily reconcealed with typical Mg-suite origin. They may crystallize from an ultra-magnesian lava with Ni contents at least twice as high as those of typical Mg-suite parent magmas. However, no such ultra-magnesian lavas or basalts containing such high-Ni forsteritic olivines have been found in previous lunar collections^{14,38}. Given our still limited understanding of lunar rock diversity, we cannot simply rule out the existence of such new type of ultra-magnesian and Ni-rich lavas. If present, the ultra-magnesian lava would have a similar genesis to the terrestrial komatiites⁴², likely being produced by extensive melting of early mantle cumulates to achieve the most primitive signature of the **Group III** olivine (Supplementary Fig. 7). The hypothetical Ni-depletion event accounting for the origin of Mg-suite magmas would not have exerted an effect, in order to prevent the ultra-magnesian lava from Ni-depletion. Further in-depth study of the CE-6 returned samples and future lunar exploration would help test the possible existence of this new type of lava on the Moon. However, we propose below an alternative possibility for the origin of the **Group III** olivine: the first olivine crystallized from the LMO.*

*Laboratory experiments and numerical simulations of LMO solidification suggest that the Mg-rich olivine is the earliest crystallized mineral from the cooling magma ocean, with Fo varying from 96.0 to 87.6, corresponding to 0 to ~50 % solidification^{6-9,15,16}. Our simulation utilizes the Earth-like Lunar Primitive Upper Mantle (LPUM) composition¹³ as the initial composition of the LMO. The Taylor Whole Moon (TWM) composition⁴³ has an apparently low Mg-number of 84 that is unable to produce olivine composition with $Fo > 94$ ^{7,9,15}. We also model the Ni content of olivine during LMO solidification following a well-established approach²⁶. The empirical Beattie-Jones model^{44,45} was employed to understand the partition of Ni between olivine and liquid: $D_{Ni}^{Ol/L} = 3.346 * D_{MgO}^{Ol/L} - 3.665$ ⁴⁴, in which the $D_{MgO}^{Ol/L} = MgO^{Ol}/MgO^L$ can be calculated at each step of LMO crystallization. Several studies estimated the initial Ni content of the LMO (also the Bulk Silicate Moon) to be depleted by a factor of about 3 ~ 4 compared to the terrestrial upper mantle, around 415 ± 105 ppm^{14,46}. Consequently, our simulation on LMO crystallization is in accordance with previous attempts^{7,26} and demonstrates that the first crystallized olivine has a Fo value of 96.2 with Ni content of 509 ± 131 ppm (Fig. 4). The Ni content of olivine increases with ongoing LMO solidification and reaches a maximum at Fo of ~92.4, after which the Ni content of olivine drops with decreasing Fo (Fig. 4). Compositions of the **Group III** olivines correspond well with the model results and thus could be of mantle origin. This inference is also supported by our additional modellings on evolution of Ni/Co ratios of olivines during LMO differentiation (Supplementary Fig. 8).*

*We acknowledge that the above two hypotheses for the origin of **Group III** olivine are not mutually*

*exclusive. Given the equilibrium crystallization of olivine at the early stage of LMO solidification and the formation of the ultra-magnesian magma through high-degree partial melting of those olivine cumulates, the olivine crystallized from the ultra-magnesian magma is expected to have similar compositions to its source component, i.e., the early LMO cumulate (Supplementary Fig. 7). That is, the **Group III** olivine resembles the first olivine crystallized from the LMO in terms of composition.”*

First, mantle rocks on Earth, and in the only recognized sample of lunar mantle material (Treiman and Semprich, 2023), do not contain euhedral or subhedral olivines, i.e. the edges of their olivine grains are irregular and not related to the orientations of the grains' crystal structures. However, the manuscript claims the opposite for the Chang'e-6 magnesian olivines:

“These forsteritic olivine grains are euhedral to subhedral ...”. So, I do not think that the Chang'e-6 magnesian olivines are pieces of mantle material – they are more likely phenocrysts from magnesian basalts. It is possible that there is a translation issue here, as the olivine grains of Figures 1A-C are anhedral, but those of Figure 1D are euhedral to subhedral.

Reply: Thank you for this comment. As we have mentioned in the above replies, it is possible that the high Fo, high Ni olivine could have crystallized from hitherto unrecognized lunar ultra-magnesian lavas. Nevertheless, we provide an alternative explanation for the origin of **Group III** olivine: the first olivine crystallized from the LMO. However, as we have revised in line 178-221, these two hypotheses on the origin of **Group III** are not mutually exclusive.

We agree with the reviewer that terrestrial mantle olivines and the one reported by the reviewer do not show perfect euhedral or subhedral structure. However, we do not necessarily agree with that early lunar cumulate olivines show similar textures as the modern terrestrial mantle olivines. Olivines crystallized during the equilibrium crystallization of the early Lunar Magma Ocean are likely to be euhedral, similar to the dunite cumulate. Later melt-related metasomatism is able to alter the olivine, though.

Second, the Ni contents of the Chang'e-6 magnesian olivines are similar to those in lunar basalts, as shown in this graph, which is Figure 1a of Longhi et al. (2010) with data for the manuscript's olivines as smaller symbols. One might reasonably expect that mantle olivine would have Ni abundances as high or higher than those from basalts. So, Ni itself does not prove a mantle origin.

Reply: Thank you for the comment. We acknowledge that the high Ni content does not necessarily prove that these olivines originated from the lunar mantle. Our simulations show that the high Fo, high Ni olivines are consistent with the composition of olivines crystallized during the early stages of LMO solidification, in agreement with the model results of Longhi et al. (2010). Nonetheless, we agree with the other possibility as you suggest: these olivines may have crystallized from an unrecognized ultra-magnesian magma possibly formed by the extensive remelting of the early LMO cumulates. We also did the modelling of extensive melting of early LMO cumulates and crystallization of olivine from this ultra-magnesian magma in Supplementary Fig. 7. Please see **Fig. 5 below**.

Overall, the two hypotheses on the origin of the high-Fo and high-Ni olivines are not mutually exclusive. Given the high degree of melting, the compositions of olivine crystallized from the ultra-magnesian magma would resemble the lunar mantle olivine, comparable with olivine from terrestrial komatiite lavas versus peridotites. Therefore, regardless of the scenario, the high-Fo, high-Ni olivines in our study can represent the composition of the early lunar mantle olivine cumulates.

Third is the question of the high Mg* of the Chang'e-6 magnesian olivines. No known lunar basalts contain olivines so magnesian, but part of the point of Chang'e-6 was to explore an area of unknown geology, and extend our understanding of the diversity of lunar rocks. Is it possible that a lunar basalt could be so magnesian as to crystallize such olivines as phenocrysts? I think the answer is yes. I base that answer on the existence of comparable lavas on Earth, the ultra-magnesian komatiite lavas of the Comondale area, South Africa (Wilson et al. 2019). These lavas contain olivine phenocrysts with Mg* to 96% (comparable to those reported here) and with up to ~2000 ppm Ni. As the Moon's mantle is inferred to contain 1/3 to 1/4 of the Ni as the Earth's mantle¹, this would suggest the possibility of a lunar ultrabasic lava with Mg* of ~96% and 500-700 ppm Ni. None has been reported so far, but I think it is possible that the high-Mg olivines reported here could be from a magma like this.

Reply: Thank you for the comment. We agree with you and make an effort to discuss this possibility in the revision. However, no such magmas have been identified on the Moon to date (Delano, 1986; Shearer et al., 2015). If present, the ultra-magnesian lava would have a similar genesis to the terrestrial komatiites, likely being produced by extensive melting of early mantle cumulates to achieve the most primitive signature of the high Fo, high Ni olivine (Fig. 5 below). Additionally, given the high degree of melting, the compositions of olivine crystallized from the ultra-magnesian magma would resemble the lunar mantle olivine, comparable with olivine from terrestrial komatiite lavas versus peridotites. Therefore, regardless of the scenario, the high-Fo, high-Ni olivines in our study can represent the composition of the early lunar mantle olivine cumulates.

Please see our revisions in Lines 178 to 221 for the detailed discussion.

Fig. 5 Thermodynamic modeling of Fo versus Ni content of olivine during equilibrium melting of the LMO cumulate pile (LPUMcp) with an initial composition of the LPUM (Longhi et al., 2006) at 50 per cent solidified from this study (blue) and Elardo et al. (2011) (orange). The squares stand for the melting degree (%). The EPMA/LA-ICPMS analyses of *Group III* olivines are shown as the blue circles/red diamonds. Error bars represent 2 σ of LA-ICPMS analysis.

Reference:

- Kohn, S., Henderson, C. & Mason, R. Element zoning trends in olivine phenocrysts from a supposed primary high-magnesian andesite: an electron-and ion-microprobe study. *Contributions to Mineralogy and Petrology* 103, 242-252 (1989).
- Fagan, A., Neal, C., Simonetti, A., Donohue, P. & O'Sullivan, K. Distinguishing between Apollo 14 impact melt and pristine mare basalt samples by geochemical and textural analyses of olivine. *Geochimica et Cosmochimica Acta* 106, 429-445 (2013).
- Zhao, D., Arai, T., Liu, L. & Ohtani, E. Seismic tomography and geochemical evidence for lunar mantle heterogeneity: comparing with Earth. *Global and Planetary Change* 90, 29-36 (2012).
- Harada, Y. et al. Strong tidal heating in an ultralow-viscosity zone at the core–mantle boundary of the Moon. *Nature Geoscience* 7, 569-572 (2014).
- Arai, T. & Maruyama, S. Formation of anorthosite on the Moon through magma ocean fractional crystallization. *Geoscience Frontiers* 8, 299-308 (2017).
- Longhi, J., Durand, S. R. & Walker, D. The pattern of Ni and Co abundances in lunar olivines. *Geochimica et Cosmochimica Acta* 74, 784-798 (2010).
- Delano, J. W. (1986). Abundances of cobalt, nickel, and volatiles in the silicate portion of the Moon. In *Origin of the Moon* (pp. 231-247).
- Shearer, C. K., Elardo, S. M., Petro, N. E., Borg, L. E. & McCubbin, F. M. Origin of the lunar highlands Mg-suite: An integrated petrology, geochemistry, chronology, and remote sensing perspective. *American Mineralogist* 100, 294-325 (2015).
- Longhi, J. Petrogenesis of picritic mare magmas: constraints on the extent of early lunar differentiation. *Geochimica et Cosmochimica Acta* 70, 5919-5934 (2006).
- Elardo, S. M., Draper, D. S. & Shearer Jr, C. K. Lunar Magma Ocean crystallization revisited: Bulk composition, early cumulate mineralogy, and the source regions of the highlands Mg-suite. *Geochimica et Cosmochimica Acta* 75, 3024-3045 (2011).

Review of Nature Communications submission 559489: “Lunar primitive mantle olivine returned by Chang’e-6” by Si-Zhang Sheng et al.

I recommend that the manuscript would be acceptable for publication, after revisions. I’ve attached a version of the submitted manuscript with a few edits to the text, a few additional comments, and some suggestions of additional references.

This manuscript reports intriguing high-Mg olivine crystals from the regolith returned by the Chang’e-6 spacecraft from the Moon’s South-Pole Aitkin Basin. The data are an important contribution to understanding the Moon, but I am not convinced that the high-Mg olivines are necessarily samples of the Moon’s mantle. The authors come to this inference because the olivine grains are very magnesian, $Mg^* = \text{molar Mg}/(\text{Mg}+\text{Fe})$ up to almost 96%, and have high Ni abundances, up to ~650 ppm. No other known lunar olivines have both these characteristics, neither basalts nor basaltic plutonic rocks. The authors show that such olivine compositions could have formed in cumulate rocks from the early Lunar Magma Ocean.

This argument is reasonable, and I have no serious problems with it. However, there is another possible interpretation that ought to be mentioned in the manuscript; an interpretation which (sadly) changes the focus and apparent importance of the olivines. It is reasonable that the magnesian nickeloan olivines crystallized from ultrabasic lavas unlike those discovered so far. This might seem far-fetched, but no more so than finding fragments of mantle olivine.

First, mantle rocks on Earth, and in the only recognized sample of lunar mantle material (Treiman and Semprich, 2023), do not contain euhedral or subhedral olivines, i.e. the edges of their olivine grains are irregular and not related to the orientations of the grains’ crystal structures. However, the manuscript claims the opposite for the Chang’e-6 magnesian olivines:

“These forsteritic olivine grains are euhedral to subhedral ...”. So, I do not think that the Chang’e-6 magnesian olivines are pieces of mantle material – they are more likely phenocrysts from magnesian basalts. It is possible that there is a translation issue here, as the olivine grains of Figures 1A-C are anhedral, but those of Figure 1D are euhedral to subhedral.

Second, the Ni contents of the Chang’e-6 magnesian olivines are similar to those in lunar basalts, as shown in this graph, which is Figure 1a of Longhi et al. (2010) with data for the manuscript’s olivines as smaller symbols. One might reasonably expect that mantle olivine would have Ni abundances as high or higher than those from basalts. So, Ni itself does not prove a mantle origin.

[REDACTED]

Third is the question of the high Mg* of the Chang’e-6 magnesian olivines. No known lunar basalts contain olivines so magnesian, but part of the point of Chang’e-6 was to explore an area of unknown geology, and extend our understanding of the diversity of lunar rocks. Is it

possible that a lunar basalt could be so magnesian as to crystallize such olivines as phenocrysts? I think the answer is yes. I base that answer on the existence of comparable lavas on Earth, the ultra-magnesian komatiite lavas of the Comondale area, South Africa (Wilson et al. 2019). These lavas contain olivine phenocrysts with Mg* to 96% (comparable to those reported here) and with up to ~2000 ppm Ni. As the Moon's mantle is inferred to contain 1/3 to 1/4 of the Ni as the Earth's mantle¹, this would suggest the possibility of a lunar ultrabasic lava with Mg* of ~96% and 500-700 ppm Ni. None has been reported so far, but I think it is possible that the high-Mg olivines reported here could be from a magma like this.

REFERENCES

- Longhi, J., Durand, S. R., & Walker, D. (2010). The pattern of Ni and Co abundances in lunar olivines. *Geochimica et Cosmochimica Acta* 74, 784-798.
- Treiman, A. H., & Semprich, J. (2023). A dunite fragment in meteorite Northwest Africa (NWA) 11421: A piece of the Moon's mantle. *American Mineralogist*, 108(12), 2182-2192.
- Wilson, A. H. (2019). The late-Paleoarchean ultra-depleted Comondale komatiites: Earth's hottest lavas and consequences for eruption. *Journal of Petrology* 60(8), 1575-1620.

END – Allan H. Treiman

¹ From the manuscript. "Several studies estimated the initial Ni content of the LMO (also the Bulk Silicate Moon) to be depleted by a factor of about 3 - 4 compared to the terrestrial upper mantle, around 415 ± 105 ppm^{45,46}."